# HRV-Guided Training for Professional Endurance Athletes: A Protocol for a Cluster-Randomized Controlled Trial

**DOI:** 10.3390/ijerph17155465

**Published:** 2020-07-29

**Authors:** María Carrasco-Poyatos, Alberto González-Quílez, Ignacio Martínez-González-Moro, Antonio Granero-Gallegos

**Affiliations:** 1Department of Education, Health and Public Administration Research Center, University of Almeria, 04120 Almeria, Spain; carrasco@ual.es; 2Department of Education, University of Almeria, 04120 Almeria, Spain; albertillo_gq@hotmail.com; 3Department of Physiotherapy, Physical Exercise and Human Performance Research Group, University of Murcia, 30001 Murcia, Spain; igmartgm@um.es

**Keywords:** HRV, endurance training, training performance, high level athletes, VO_2max_, running

## Abstract

Physiological training responses depend on sympathetic (SNS) and parasympathetic nervous system (PNS) balance. This activity can be measured using heart rate variability (HRV). Such a measurement method can favor individualized training planning to improve athletes’ performance. Recently, HRV-guided training has been implemented both on professional and amateur sportsmen and sportswomen with varied results. There is a dearth of studies involving professional endurance athletes following a defined HRV-guided training protocol. The objectives of the proposed protocol are: (i) to determine changes in the performance of high-level athletes after following an HRV-guided or a traditional training period and (ii) to determine differences in the athletes’ performance after following both training protocols. This will be a 12-week cluster-randomized controlled protocol in which professional athletes will be assigned to an HRV-based training group (HRV-G) or a traditional-based training group (TRAD-G). TRAD-G will train according to a predefined training program. HRV-G training will depend on the athletes’ daily HRV. The maximal oxygen uptake (VO_2max_) attained in an incremental treadmill test will be considered as the primary outcome. It is expected that this HRV-guided training protocol will improve functional performance in the high-level athletes, achieving better results than a traditional training method, and thus providing a good strategy for coaches of high-level athletes.

## 1. Introduction

It is known that training is essential for improving physical performance [1] and that optimizing training for performance improvement in athletes is an important area of research within exercise physiology and sports medicine [2,3,4]. In this regard, different training methods for performance improvement have been tried and tested, such as intensified training [2,5,6] and submaximal tests [7]. However, it is also recognized that using the same standardized training program for a group of athletes can provoke a wide range of reactions in terms of performance and physiological adaptations [8,9].

As stated by Schmitt, Willis, Fardel, Coulmy, and Millet [10], an important component of the interindividual variability in physiological responses to standardized training is related to the balance between the parasympathetic (PNS) and sympathetic (SNS) activity of the autonomic nervous system (ANS) [11,12]. Heart rate variability (HRV) is one of the indicators that allows the noninvasive study of autonomic nervous system activity in its sympathetic and parasympathetic branches [13,14]. HRV is understood as the variation in the time interval between two consecutive heartbeats. It is obtained by calculating the time interval that elapses between two consecutive R waves (i.e., RR interval fluctuation) on an electrocardiogram (ECG) [15]. The period between beats is not constant; consequently, high HRV values are associated with an efficient ANS, which promotes behavioral adaptation and cognitive flexibility during stress [16], whilst low HRV values are indicative of an inefficient ANS, resulting in maladaptive responses to stress and perceived threats [15]. Furthermore, HRV is considered to be an indicator of cardiovascular health level [17].

Given that the SNS is responsible for changes in heart rate (HR) due to stress, and that the HR is one of the first parameters used to control the body’s functional capacity [18], HRV analysis has been established as a useful method for assessing the heart’s ability to adapt to both endogenous and exogenous loads [19], and can be used for the individual assessment of responses to training loads. Indeed, in recent years, HRV has been used to analyze these imbalances between SNS and PNS in athletes [20] and to evaluate different aspects related to training [21] such as exercise intensity and duration [22], recovery and overtraining [23], training load [24] or psychophysiological profiles [25].

The control of training based on HRV, as an indicator of the precompetitive physical and psychological state in athletes, enables coaches and scientists to use these HRV records to adapt the recovery and training loads to each athlete in search of a better sports result. As indicated by Ortigosa-Márquez, Reigal, Carranque, and Hernández-Mendo [26], high HRV values indicate more parasympathetic than sympathetic activation in an athlete and, therefore, better recovery and preparation for dealing with high-intensity training sessions.

Traditionally, HRV has been measured with ECG [14]. One of the ways of quantifying HRV is through rMSSD (the root mean square of successive differences between adjacent RR intervals) [26] since it is a temporal statistical parameter that reports those variations occurring over the short term between RR intervals [27] and it is used to observe the influence of the SNP on the cardiovascular system [18]. Currently, there are other validated tools for determining HRV that facilitate measurement, such as the Kubios HRV, Elite HRV, Mobile Lab and HRV4Training applications (apps).

In recent years, experimental studies have been carried out evaluating HRV-guided training in endurance athletes. These studies have been conducted both on elite athletes in sports such as cycling [24] and skiing [10], and with amateur endurance athletes [28,29,30,31]. One should take into account the scarcity of studies that have been published to date, especially on elite-level endurance athletes, as well as the absence of a common protocol to follow in this type of research.

The protocol proposed in the present study contributes to the scientific literature in this field in several ways: (i) it proposes research focused on elite athletes, a sample population for which only two experimental studies have been carried out to date; (ii) the protocol is intended to research endurance runners, for which we are not aware of any research having been carried out on this type of sample and with these characteristics at the international level. Therefore, it is a novel study aiming to provide empirical support for HRV-guided training in long-distance runners, adapting daily training to the physiological responses of each individual athlete. The performance of these athletes could be compared to that of another group of long-distance runners who carry out traditional training over the same period of time.

Until just a few years ago, conducting research of this type, involving daily HRV measurements on each athlete and then adapting training based on these data, was only possible with the collaboration of high-cost laboratories. This study tests the use of noninvasive, commercial, low-cost and publicly accessible technology to evaluate the physiological responses obtained by adapting training to HRV.

Based on everything described above, we hypothesize that HRV-guided training will: (i) improve functional performance in high-level athletes and (ii) produce better performance results than a traditional training method. The objectives of the proposed protocol are: (i) to determine changes in the performance of high-level athletes after following an HRV-guided or a traditional training period, and (ii) to determine differences in the athletes’ performance after following both training protocols.

This will be a 12-week cluster-randomized controlled trial protocol in which professional athletes are assigned to either an HRV-based training group (HRV-G) or a traditional-based training group (TRAD-G). A block randomization method will be chosen to randomly assign participants to interventions in equally sized sample groups. This protocol has been designed following the Standard Protocol Items: Recommendations for Interventional Trials (SPIRIT) Statement [32]. To describe the intervention, the TIDieR (Template for Intervention Description and Replication) checklist by Hoffmann et al. [33] has been used.

## 2. Materials and Methods

### 2.1. Study Setting

To detect an intervention-related effect in professional athletes, other studies with similar protocols [24,31] compared athletes from two clubs or associations. Similarly, our sample will comprise athletes from two sport institutions in Almería (Spain): the C.D. Atletas de Almería, based in the city of Almería (Spain) and the Asociación Espeleológica Velezana, based in Vélez Rubio (Spain).

### 2.2. Eligibility Criteria

The inclusion criteria for participating in the program will be: (i) to belong to the Spanish Athletics Federation; (ii) to have been training and competing in Spanish Athletics Federation competitions for at least two years; and iii) to be in the first third of the classification for the last five races of the previous season. Regarding the exclusion criteria, the following will be taken into account: (i) having cardiovascular pathologies, abnormal blood pressure parameters or diagnosed respiratory problems; (ii) being treated for psychological problems, or regularly taking a drug(s) that has a direct or indirect effect on the nervous system (e.g., anxiolytics, antidepressants or neuroleptics); (iii) substance use that is not permitted by the International Association of Athletics Federations (IAAF); (iv) occasional consumption of medication for a disease related to the cardiorespiratory system (e.g., influenza) that might alter performance and (v) not performing at least 90% of the workouts during the intervention.

The trial steering members will be responsible for checking that the subjects interviewed meet the inclusion criteria. The Spanish Athletics Federation’s medical team will certify that the subjects do not meet any of the exclusion criteria. After being informed of the study design and potential risks, all athletes will sign a written informed consent document. The model consent form is shown in Appendix A.

### 2.3. Interventions

Based on the methodology used by Javaloyes et al. [24] and Vesterinen et al. [31], the intervention will be divided into two training periods for both study groups (HRV-G and TRAD-G): a four-week preparation period (PR) and an eight-week training period (TR). Both will maintain the weekly training volume. The training carried out will mainly be running. The PR period will be common to both groups and will be a familiarization phase for the training sessions and their intensities. During this period, the training intensity will gradually increase for the first three weeks and then decrease in the fourth week. This will mean three weeks of overloading and one week of recovery (3:1). The training to be carried out by the athletes is presented in more detail in Table 1. In the TR period, each group will carry out the corresponding intervention. The TRAD-G group will train according to a predefined training program, which will include sessions carried out at low intensity (approximately 50% of the total), and other sessions of moderate and high intensity, with a structure similar to that carried out during the PR period (Table 2). The training prescribed to the HRV-G group will depend on the subjects’ HRV, in accordance with authors such as Javaloyes et al. [24], Kiviniemi, Hautala, Kinnunen, and Tulppo [34], and Lamberts, Swart, Noakes, and Lambert [35].

To quantify the HRV, a Smartphone application known as “HRV4Training” (see http://www.hrv4training.com/) will be used. This tool has been validated by Plews et al. [36], showing a low typical estimate error (CV% (90% CI) = 3.8 (3.1; 5.0)) and a clear electrocardiographical correlation (r = 1.00 (1.00; 1.00)). It provides the root mean sum of the successive differences between R – R intervals (rMSSD) data using photoplethysmography. rMSSD is more suitable and reliable than other indexes [13,37]; nonetheless, the HRV data will be transformed by taking the natural logarithm, thus allowing parametric statistical comparisons that assume a normal distribution. In this way, a 7-day rolling average will be calculated (LnrMSSD_7-d_). The PR period will be used as a standardized phase to obtain the baseline LnrMSSD_7-d_ and its range of normality (upper and lower limits). Following the indications of Plews, Laursen, Kilding, and Buchheit [38], this will be calculated as the mean ± 0.5 × SD. During the TR period, the LnrMSSD_7-d_ will be calculated daily in order to adapt the training prescribed to the HRV-G athletes. Moreover, the range of normality will be updated weekly. If the LnrMSSD_7-d_ is within the range of normality, the athletes will perform a moderate or high-intensity session. If the weekly LnrMSSD_7-d_ average falls below the normal range, a low intensity workout or rest will be undertaken. Athletes will perform a maximum of two consecutive sessions of moderate or high intensity; likewise, they will not accumulate more than two consecutive rest sessions. The modified scheme of Kiviniemi et al. [34] presented in Figure 1 will be followed.

In accordance with Javaloyes et al. [24], all participants will be instructed to measure their HRV data at home each morning after waking up and emptying their bladders. They will be instructed to lie in a supine position and not perform any further activity during the recordings. Data will be recorded over a 60-s period e.g., [36]. The daily control and recording of the rMSSD, as well as the LnrMSSD_7-d_ calculation used to prescribe the training of the HRV-G athletes, will always be carried out by the trial steering members. These members will receive the information from each athlete via phone or email and, in turn, will inform the HRV-G coach of the training intensity corresponding to each athlete. This procedure will also serve as a strategy for maintaining and monitoring the athletes’ adherence to the training programs. Concomitant care, or any other intervention, will not be allowed during the trial for either the HRV-G or the TRAD-G. Athletes from both groups will carry out the training in their usual location.

### 2.4. Outcomes

The primary outcome of this study will be the maximal oxygen uptake (VO_2max_) obtained in an incremental treadmill test. The secondary outcomes will be: the maximal speed in m/s, maximal heart rate, respiratory exchange ratio, ventilatory thresholds (VT1 and VT2) and their derived speed, heart rate, respiratory exchange ratio and VO_2_ obtained in the incremental treadmill test. Other measurements considered as secondary outcomes will be: the time, speed, heart rate, rating of perceived exertion (RPE) and lactate in the 3000 m running test. Body composition and rMSSD will be considered as other variables.

Measurements will be taken before and after the training period, which will correspond to weeks 5 (pretest) and 12 (post-test). Over the assessment weeks, care will be taken that participants do not carry out any high-intensity training sessions. Each assessment week will consist of two testing sessions with a 48-h recovery period. The first testing session will include maximal graded exercise test and body composition measurements. In the second testing session, athletes will perform a 3000 m running test. The rMSSD will be measured daily, as explained in the intervention section.

The incremental treadmill test will be performed by the Physical Exercise and Human Performance Research Group at the University of Murcia (Spain). This is a more objective way of determining physical fitness and represents the maximal performance capacity of an individual [39]. First, with the athlete in the supine position, a cardiovascular examination will be carried out at rest by means of cardiac auscultation, blood pressure and an electrocardiogram (ECG). The electrodes for recording the ECG and heart rate will be kept in place throughout the test. The Cardioline Cube^®^ electrocardiograph will be used. To perform the incremental treadmill test, the Runner srl (Cavezzo Italy) treadmill will be used, as it was in other studies such as Ballesta-García, Martínez-González-Moro, Ramos-Campo, and Carrasco-Poyatos [40]. Similar to other studies, such as Nuuttila et al. [30] or Vesterinen et al. [31], a prior 2-min aerobic warm up will be performed at 6 km/h. The test itself will start at a velocity of 7 km/h. The speed will be increased by 0.1 km/h every 6 s. The incline will remain at 1% throughout the test. The athletes will be encouraged to perform at maximum effort. The test will end when the subject can no longer run; the subject will indicate this with a hand gesture. The recovery phase will then begin at 4km/h for 3 min followed by rest for a further 2 min. The tests will be considered maximal and valid when the theoretical heart rate (220-age) exceeds 85% and the respiratory exchange ratio (RER) is greater than 1.15 [41]. During the stress test, the subjects will breathe through a mask connected to a gas analyzer (Metalyzer 3b^®^, Cortex, Leipzig, Germany). All gas exchange parameters will be measured breath-by-breath and averaged every 30 s. The VO_2max_ will be defined as the oxygen consumption plateau [42]. The aerobic (VT1) and anaerobic (VT2) thresholds will be determined. Before each test, the gas analyzer will be manually calibrated. The test’s maximal speed (V_max_), maximal heart rate (HR_max_), and respiratory exchange ratio (RER) will be recorded. The V_max_ or HR_max_ will be defined as the highest speed, or heart rate, reached for a finished stage. The speed, heart rate, respiratory exchange ratio and VO_2_ at each ventilatory threshold will also be recorded as V_VT1,_ V_VT2_, HR_VT1_, HR_VT2_, RER_VT1_, RER_VT2_, VO_2VT1_, and VO_2VT2_, respectively. All tests will be carried out under similar environmental conditions (an ambient temperature of 20–22 degrees).

As in other studies [30,43], the 3000 m running test will be conducted individually on a 400 m outdoor running track. Participants will be instructed to run at their maximum speed. Before the test, a 15 min standardized aerobic warm-up will be performed, consisting of running at a low to moderate intensity. Capillary blood samples (5 μL) for blood lactate concentration analysis will be taken from the fingertip using a Scout+ analyzer (SensLab GmbH, Leipzig, Germany). Lactate is considered a useful indicator for measuring the metabolic cost and intensity of effort in aerobic-anaerobic sports [44]. Following Ribas [45], it will be considered in this test to relate it to running intensity. Lactate samples will be taken at four different points in time, in accordance with Rodríguez and Valero [46]: (i) just before the test (Lactate_pre_), (ii) just after the test (Lactate_post_), (iii) 3 min after the test (Lactate_post3_) and iv) 5 min after the test (Lactate_post5_). Other variables, such as heart rate, time, speed, and rated perceived exertion (RPE), will also be measured. Heart rate will be recorded at five different points in time: (i) just before the test (HR_pre_), (ii) just after the test (HR_post_), (iii) 1 min after the test (HR_post1_), (iv) 3 min after the test (HR_post3_) and v) 5 min after the test (HR_post5_). The time will be recorded every 1000 m at three different points in time: (i) after running 1000 m, (ii) after running 2000 m and (iii) after running 3000 m (right at the end of the test). The speed will be calculated from these three points using the formula: speed = distance in m/time in seconds. The RPE will be measured at the end of the test using the modified Börg CR-10 scale of perceived exertion [47]. According to the authors, a 0 rating corresponds to rest; a 3 rating to moderate intensity; a 5 rating to hard intensity; a 7 rating to very hard intensity; and a 10 rating to maximal intensity. This tool has recently been determined as a stand-alone method for training load monitoring purposes in several sports and physical activities with men and women in different age categories (children, adolescents and adults) at various expertise levels [48].

Body composition will be analyzed just before the treadmill test using the InBody120 analyzer (Biospace Co. Ltd., Seoul, South Korea). Height will be measured using a measuring rod (Seca 213), and the body mass index (BMI) will be calculated according to the formula: BMI = kg/m^2^.

The time schedule for enrolment, interventions and assessments is shown in Figure 2.

### 2.5. Sample Size and Power

Calculations to establish the sample size will be performed using RStudio 3.15.0 software (PBC, Boston, USA). The significance level will be set at *p* ≤ 0.05. According to the mean standard deviation established for VO_2max_ in a previous study [31] (SD = 1.5 mL/kg/min) and an estimated error (*d*) of 1.1, a valid sample size providing a 95% confidence interval (CI) in each group will be 7 (*n* = *CI*^2^ × *d*^2^/*SD*^2^). Thus, a final sample size of 7 for each group will provide a power of 93% if between and within a variance of 2.

### 2.6. Recruitment

Each club or association involved in athletics in Almería (Spain) will be screened to identify the percentage of high-level or professionally federated athletes. When there are at least 7 high-level or professionally federated athletes, the club/association officers will be contacted by telephone to inform them of the study objective. Once they agree, an informative talk will be carried out with the athletes and the coach to inform them of the study objective, the time period in which it will take place, and the required commitment by the athletes to measure their daily HRV according to the established protocol, to attend the pre and post-test sessions and to attend at least 90% of the training sessions. The coaches will be informed of the required commitment to adapt each athlete’s training session to the daily HRV score if their club is randomized into the HRV-G group. If they agree to participate, then they will have to sign the written consent and meet the eligibility criteria necessary to be recruited into the study. The recruitment process will be conducted by the trial steering members.

### 2.7. Allocation and Blinding

A block randomization method will be used to allocate participants to the groups, which will contain equal sample sizes. The block size will be determined by the data monitoring committee according to the statistical power provided. Blocks will be chosen randomly by tossing a coin to determine the participants’ assignment into the groups. This procedure will be carried out by the data monitoring committee. The athletes and the data monitoring committee will be blinded to the exercise group assignment.

### 2.8. Data Analysis

Data will be analyzed using Jamovi (Jamovi Project 2018, version 0.9.1.7, Sydney, Australia) and RStudio 3.15.0 software (PBC, Boston, MA, USA). Prior to data analysis, the Shapiro Wilk test and the Levene test will be performed to determine the normal distribution of the variables and the homogeneity of variance. Descriptive data will be reported as mean ± SD and range. All the data will be analyzed based on the intention-to-treat principle (last observation carried forward). If the sample is normally distributed, Student’s t-test will be calculated to compare variables before and after the intervention. For a variable to be considered as having a normal distribution, 95% of values will have to be within two standard deviations of the mean. If the sample is nonparametric, the U-Mann Whitney test will be used to compare variables before and after the intervention. The standardized mean differences (Cohen’s effect size) will be calculated together with the 95% confidence intervals [49]. The effect sizes (ES) will be calculated using Cohen’s d [49]. The relationship between variables will be assessed using the Pearson r correlation coefficient. If r is higher than 0.7, the determination coefficient (r^2^) will be used to determine the percentage of Y variation with regard to the X variation. Significance will be accepted at *p* ≤ 0.05.

### 2.9. Monitoring

A data monitoring committee will be set up during the study recruitment period. Interim analyses will be supplied to the committee in strict confidence, together with any other analyses that the committee may request. Based on the data monitoring committee’s advice, the trial steering members will decide whether or not to modify the trial intake.

In our study, an adverse event will be defined as any untoward medical occurrence in a subject without regard to the possibility of a causal relationship. Adverse events will be collected after the subject has provided consent and enrolled in the study. If a subject experiences an adverse event after the informed consent document is signed (entry) but the subject has not started to receive study intervention, the event will be reported as not being related to the study’s exercise program. For this study, the following will be considered serious adverse events: severe or permanent disability, use of prohibited substances and any other significant hazard as determined by the study members. Serious adverse events occurring after a subject has stopped participating in the study will not be reported unless the researchers feel that the event may have been caused by the study protocol procedure.

### 2.10. Ethics and Dissemination

This protocol, the informed consent template contained in Appendix A and other requested documents (if any) will be reviewed and approved by the Bioethical Committee at the University of Almería with respect to the scientific content and compliance with applicable research and human subject regulations. Following initial review and approval, this protocol will be reviewed by the researcher at least once a year at Clinicaltrials.org, where it is registered with the ID: NCT04150952.

Any protocol modifications which might have an impact on conducting the study, potentially benefit a subject or affect a subject’s safety, or change the study objectives, study design, subject population, sample sizes, study procedures, along with significant administrative issues, will require a formal amendment to the protocol. Such an amendment will be agreed upon by the Bioethical Committee at the University of Almería prior to implementation, and the clubs/associations enrolled will be notified. Administrative changes to the protocol that are minor corrections, and/or clarifications having no effect on the way the study is to be conducted, will be agreed upon by the researchers and documented in a memorandum. The Bioethical Committee at the University of Almería may be notified of the administrative changes.

All study-related information will be stored securely at the study site. All records that contain names or other personal identifiers, such as locator forms and informed consent forms, will be stored separately from the study records and identified by a code number. Forms, lists, logbooks, appointment books and any other listings that link participant ID numbers to other identifying information will be stored in a separate, locked file.

## 3. Discussion

This protocol describes the rationale, design, and methods of an HRV-guided training design for professional endurance athletes. It will allow accomplishment of a randomized controlled intervention to determine changes in the performance of high-level athletes after following an HRV-guided or a traditional training period, Moreover, the differences in the athletes’ performance after following both training protocols will be determined. To design this protocol with professional endurance athletes, the guidelines described in Kiviniemi et al. [34] have been followed. This procedure has also been adapted in other professional sports such as cycling [24,50] and skiing [10], as well as to amateur endurance athletes [28,29,30,31].

This is the first time that this kind of protocol will be applied in endurance elite athletes. After its implementation, we expect that both high-level athletes groups (HRV-G and TRAD-G) improve: (i) VO_2max_ and other secondary outcomes measured in the treadmill test (the maximal speed in m/s, maximal heart rate, respiratory exchange ratio, or ventilatory thresholds), (ii) the time, speed, heart rate, rating of perceived exertion (RPE) and lactate in the 3000 m running test. Additionally, HRV-G will be better regarding performance results than the TRAD-G. These findings will suggest that training guidance balancing the sympathetic and parasympathetic autonomic nervous system leads to greater athletic performance in endurance athletes compared to standardized prescribed training. This is relevant for training optimization and for minimizing overuse and reducing injury risk.

## 4. Conclusions

Experimental research conducted in recent years shows that improvements in variables related with athletes’ performance (e.g., VO_2max_) can be obtained through HRV-guided training. However, accordingly to these studies, results do not allow a consensus to be established regarding the performance benefits of HRV-guided training for endurance athletes.

From studies carried out until now, this article describes a novel protocol to conduct a randomized controlled trial with endurance athletes. So far, no other HRV-guided training research has been conducted with these types of professional athletes. Besides, this protocol proposes to use emergent technologies in the training and the research fields, such as smartphone applications; in this case, HRV4training, an app scientifically validated that allows calculation of the daily HRV measurement for each athlete. Although more research is needed, the implementation of the protocol described here will contribute to this scientific field of study.

## Figures and Tables

**Figure 1 ijerph-17-05465-f001:**
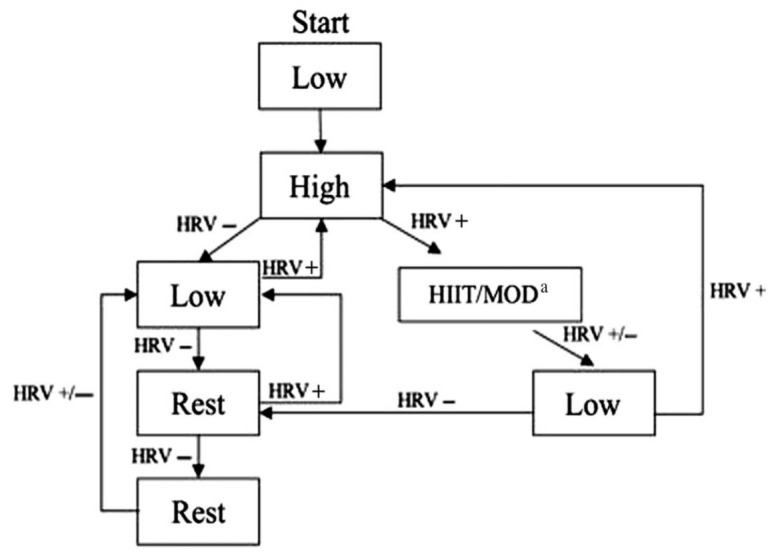
HRV-guided training schema. Modified from Kiviniemi et al. [34]. Note: When LnrMSSD7-d remained inside their normal range, high-intensity or moderate-intensity training sessions were prescribed. If LnrMSSD7-d fell outside their normal range (below), low intensity or rest were prescribed. HIIT/MOD = high/moderate-intensity interval training; HRV = heart rate variability; LnrMSSD7-d = 7-day rolling average of the natural logarithm of the root-mean-squared differences of successive RR intervals.

**Figure 2 ijerph-17-05465-f002:**
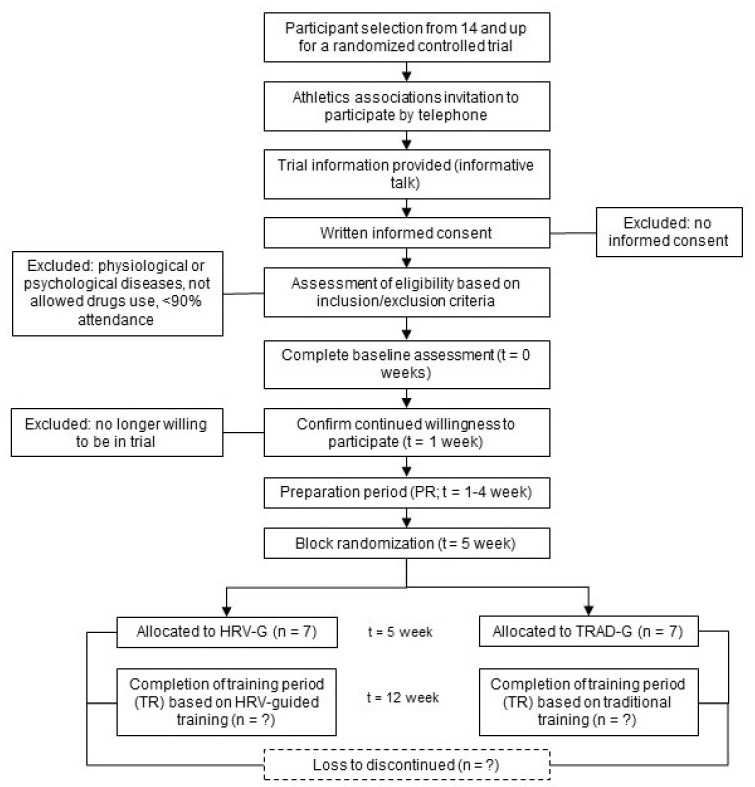
Schedule of enrolment, interventions and assessment. Note: HRV-G = heart rate variability-based training group; TRAD-G = traditional based training group; PR = preparation period; TR = training period.

**Table 1 ijerph-17-05465-t001:** Periodization and training distribution for the heart rate variability group (HRV-G) and traditional-based training group (TRAD-G) during the preparation period (PR).

Weeks	High Intensity		Moderate Intensity	Low Intensity
1			90 min between VT1 and VT2	3–4 sessions between 30 and 35 min below VT1
2		4 × 12 min > VT2/3-min rest	90 min between VT1 and VT2	2–3 sessions between 30 and 35 min below VT1
3	50 min at VT2	4 × 12 min > VT2/3-min rest	90 min between VT1 and VT2	2–3 sessions between 30 and 35 min below VT1
4				3–4 sessions between 30 and 35 min below VT1

Note: VT1 = first ventilatory threshold; VT2 = second ventilatory threshold. High-intensity and moderate-intensity sessions will be performed with a 15- to 20-min warm up and 20 min of cooling down.

**Table 2 ijerph-17-05465-t002:** Periodization and training distribution for TRAD-G during training period (TR).

Weeks	High Intensity		Moderate Intensity	Low Intensity
5	50 min at VT2			3–4 sessions between 30 and 35 min below VT1
6		4 × 12 min > VT2/3-min rest	90 min between VT1 and VT2	2–3 sessions between 30 and 35 min below VT1
7	50 min at VT2	4 × 12 min > VT2/3-min rest	90 min between VT1 and VT2	2–3 sessions between 30 and 35 min below VT1
8				3–4 sessions between 30 and 35 min below VT1
9	50 min al VT2			3–4 sessions between 30 and 35 min below VT1
10			90 min between VT1 and VT2	2–3 sessions between 30 and 35 min below VT1
11	50 min al VT2	4 × 12 min > VT2/3-min rest	90 min between VT1 and VT2	2–3 sessions between 30 and 35 min below VT1
12				3–4 sessions between 30 and 35 min below VT1

Note: VT1 = first ventilatory threshold; VT2 = second ventilatory threshold. High-intensity and moderate-intensity sessions will be performed with a 15- to 20-min warm up and 20 min of cooling down. Approximately 50% of the total sessions will be at low intensity.

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
