# Peer review of "HRV-Guided Training for Professional Endurance Athletes: A Protocol for a Cluster-Randomized Controlled Trial"

_ijerph, 2020, doi:10.3390/ijerph17155465_

Round 1
Reviewer 1 Report
Overall, the project is well outlined and clearly written.
The idea of ​​controlling training through the assessment of physiological factors makes perfect sense and is widely used in an attempt to improve performance.
The specific use of HRV or HRV4Training in training control, in my opinion, should be applied with reservations. Although HRV can measure the activity of the nervous system during effort and recovery, it is necessary to consistently prove that it accurately measures the fatigue state of the organism / individual.
That is, do we guarantee that HRV validly expresses the state of fatigue? If we think so, the validation of the protocol suggested by the authors makes perfect sense. However, although HRV can measure the complexity of nervous system activity, it seems to me an exaggeration to think that it can express, in a single value, the general state of fatigue.
As I understand it, the protocol will measure the maximum oxygen consumption (VO2max) to study the evolution of performance. In this regard, we must take into account that VO2max is not the physiological parameter most correlated with performance in medium or long distance running. The longitudinal assessment of performance could be more rigorous if we evaluate the athlete in a given run / distance.
Despite some limitations in its application, this training control procedure can provide important indications that allow training to be adjusted over time.
Author Response
Thank you for providing us with positive feedback and constructive comments regarding our submission to the International Journal of Environmental Research and Public Health entitled, “HRV-guided training for professional endurance athletes: a protocol for a cluster-randomized controlled trial” (ijerph-843333) and for inviting us to submit a revised version. First, we apologize for any confusion among reviewers for submitting the manuscript as an "article" rather than a "protocol".
Comments.- Overall, the project is well outlined and clearly written.
The idea of ​​controlling training through the assessment of physiological factors makes perfect sense and is widely used in an attempt to improve performance.
The specific use of HRV or HRV4Training in training control, in my opinion, should be applied with reservations. Although HRV can measure the activity of the nervous system during effort and recovery, it is necessary to consistently prove that it accurately measures the fatigue state of the organism / individual. That is, do we guarantee that HRV validly expresses the state of fatigue? If we think so, the validation of the protocol suggested by the authors makes perfect sense. However, although HRV can measure the complexity of nervous system activity, it seems to be an exaggeration to think that it can express, in a single value, the general state of fatigue. As I understand it, the protocol will measure the maximum oxygen consumption (VO2max) to study the evolution of performance. In this regard, we must take into account that VO2max is not the physiological parameter most correlated with performance in medium or long distance running. The longitudinal assessment of performance could be more rigorous if we evaluate the athlete in a given run / distance. Despite some limitations in its application, this training control procedure can provide important indications that allow training to be adjusted over time.
- Response: We appreciate the comments of the reviewer about our work. Regarding comments, we can guarantee that the instrument to measure HRV included in this study (HRV4training) is validated (please, see: Plews, Scott, Altini, Wood, Kilding, Laursen, 2017) and it has recently been used in the literature and published in prestigious journals (e.g. Flatt, Esco, & Nakamura, 2017; Javaloyes, Sarabia, Lamberts, & Moya-Ramon, 2019; Javaloyes, Sarabia, Lamberts, Plews, & Moya-Ramon, 2020). On the other hand, the VO2max will be assessed to measure the improvement of performance in the longitudinal assessment, but the performance of the athletes is also assessed in a given run/distance (3000m test) (see section 2.4 Outcomes).
References:
Flatt, A. A., Esco, M. R., y Nakamura, F, Y. (2017). Individual heart rate variability responses to preseason training in high level female soccer players. Journal of Strength and Conditioning Research 31(2), 531-538. Doi: 10.1519/JSC.0000000000001482
Javaloyes A.; Sarabia J. M.; Lamberts, R. P.; Plews, D.; Moya-Ramon, M. Training Prescription Guided by Heart Rate Variability Vs. Block Periodization in Well-Trained Cyclists. J Strength and Conditioning Research 2020 34(6), 1511–1518. https://doi.org/10.1519/JSC.0000000000003337
Plews D.J.; Scott B.; Altini M.; Wood M.; Kilding A.E.; Laursen P.B. Comparison of Heart-Rate-Variability Recording With Smartphone Photoplethysmography, Polar H7 Chest Strap, and Electrocardiography. Int J Sports Physiol Perform 2017, 12(10), 1324–8. https://doi.org/10.1123/ijspp.2016-0668
Reviewer 2 Report
It is not finished, the main parts (results, discussion, conclusions) of this document are missing. The authors have just written in this work what they want to do in the future. The objectives of this study are unclear. Did the authors want to confirm their methodology? I'm confused after reading this paper. I was rather expecting the results of the presented HRV training, but this study only presents plans for the future of the authors. It cannot be published as an original paper - it's just a training protocol proposal. The usefulness of this article (training method) cannot be verified and evaluated without results. This paper is not finished, there are no results, discussion and conclusions. I recommend that this study be completed and published with the results, discussion and conclusions. In the presented form it does not meet the standards of scientific work. The paper should be rejected.
Author Response
Thank you for providing us with positive feedback and constructive comments regarding our submission to the International Journal of Environmental Research and Public Health entitled, “HRV-guided training for professional endurance athletes: a protocol for a cluster-randomized controlled trial” (ijerph-843333) and for inviting us to submit a revised version. First, we apologize for any confusion among reviewers for submitting the manuscript as an "article" rather than a "protocol".
Comments.- It is not finished, the main parts (results, discussion, conclusions) of this document are missing. The authors have just written in this work what they want to do in the future. The objectives of this study are unclear. Did the authors want to confirm their methodology? I'm confused after reading this paper. I was rather expecting the results of the presented HRV training, but this study only presents plans for the future of the authors. It cannot be published as an original paper - it's just a training protocol proposal. The usefulness of this article (training method) cannot be verified and evaluated without results. This paper is not finished, there are no results, discussion and conclusions. I recommend that this study be completed and published with the results, discussion and conclusions. In the presented form it does not meet the standards of scientific work. The paper should be rejected.
- Response: Indeed, the reviewer is right. The manuscript is a protocol for a cluster-randomized controlled trial and we should have submitted the manuscript as a “protocol” rather than as an “article”. However, the reviewer can check that the protocol has been designed following the most rigorous rules and recommendations of quality for protocols: Recommendations for Interventional Trials (SPIRIT) Statement (Chan, Tetzlaff, Gotzsche, Altman, Mann, Berlin, Dickersin, Hrobjartsson, Schulz, Parulekar, Krleza-Jeric, Laupacis, Moher, 2013), and the TIDieR (Template for Intervention Description and Replication) checklist by Hoffmann et al. (Hoffmann, Glasziou, Boutron, Milne, Perera, Moher, Altman, Barbour, Macdonald, Johnston, Lamb, Dixon-Woods, McCulloch, Wyatt, Chan, Michie, 2014. It was pointed out at the end of the introduction and method sections. For this reason, as the SPIRIT and TIDIER suggest, the structure of our work has no results, discusión, or conclusions. However, it can be considered as an original paper, as other protocols published in the International Journal of Environmental and Public Health.
References:
Chan A-W.; Tetzlaff J.M.; Gotzsche P.C.; Altman D.G.; Mann H.; Berlin J.A.; Dickersin, K.; Hrobjartsson, A.; Schulz, K.F.; Parulekar, W.R.; Krleza-Jeric, K.; Laupacis, A.; Moher, D. SPIRIT 2013 explanation and elaboration: guidance for protocols of clinical trials. BMJ 2013 346(jan08 15), e7586–e7586. Available from: http://www.bmj.com/cgi/doi/10.1136/bmj.e7586
Hoffmann T.C.; Glasziou P.P.; Boutron I.; Milne R.; Perera R.; Moher D.; Altman D.G.; Barbour V.; Macdonald H.; Johnston M.; Lamb S.E.; Dixon-Woods M.; McCulloch P.; Wyatt J.C.; Chan A.-W.; Michie S. Better reporting of interventions: template for intervention description and replication (TIDieR) checklist and guide. BMJ 2014, 348, g1687–g1687. https://doi.org/10.1136/bmj.g1687
Reviewer 3 Report
The authors have described the protocol of 4 weeks of familiarization training and 8 weeks divided into two groups: HRV-Guided and Traditional-guided for a total of 12 weeks. The variables that had to be measured come from an incremental treadmill test and a 3000m test with lactate measurements. The article is easily understandable and English writing is, in my opinion, very good.
Training programs are well described for both PR and training sessions in Tables 1 and 2.
I have some concerns regarding this article:
Why isn't there a results, discussion nor conclusions section? Is this supposed to be an error or the authors just aim to define a training and testing protocol? If it is a mistake, author must submit the results, discussion and conclusion's section.
Is the speleology association an athletics club as well? Are the members of this association supposed to be professional endurance athletes?
154: Are you meaning the root mean square of successive differences between R-R intervals?
172: Authors wrote the word outside twice
179: Authors must present some studies where a period of 60-s for HRV measurements has been demonstrated to give reliable information rather than measuring for 5 minutes.
189: do authors mean maximal aerobic speed rather than maximal speed?
Author Response
Thank you for providing us with positive feedback and constructive comments regarding our submission to the International Journal of Environmental Research and Public Health entitled, “HRV-guided training for professional endurance athletes: a protocol for a cluster-randomized controlled trial” (ijerph-843333) and for inviting us to submit a revised version. First, we apologize for any confusion among reviewers for submitting the manuscript as an "article" rather than a "protocol".
Comment 1.- The authors have described the protocol of 4 weeks of familiarization training and 8 weeks divided into two groups: HRV-Guided and Traditional-guided for a total of 12 weeks. The variables that had to be measured come from an incremental treadmill test and a 3000m test with lactate measurements. The article is easily understandable and English writing is, in my opinion, very good.
- Response: We appreciate this comment, thank you so much.
Comment 2.- Why isn't there a results, discussion nor conclusions section? Is this supposed to be an error or the authors just aim to define a training and testing protocol? If it is a mistake, author must submit the results, discussion and conclusion's section.
- Response: The manuscript is a protocol for a cluster-randomized controlled trial and we should have submitted the manuscript as a “protocol” rather than as an “article”. However, the reviewer can check that the protocol has been designed following the most rigorous rules and recommendations of quality for protocols: Recommendations for Interventional Trials (SPIRIT) Statement (Chan, Tetzlaff, Gotzsche, Altman, Mann, Berlin, Dickersin, Hrobjartsson, Schulz, Parulekar, Krleza-Jeric, Laupacis, Moher, 2013), and the TIDieR (Template for Intervention Description and Replication) checklist by Hoffmann et al. (Hoffmann, Glasziou, Boutron, Milne, Perera, Moher, Altman, Barbour, Macdonald, Johnston, Lamb, Dixon-Woods, McCulloch, Wyatt, Chan, Michie, 2014. It was pointed out at the end of the introduction and method sections. For this reason, as the SPIRIT and TIDIER suggest, the structure of our work has no results, discusión, or conclusions. However, it can be considered as an original paper, as other protocols published in the International Journal of Environmental and Public Health.
Comment 3.- Is the speleology association an athletics club as well? Are the members of this association supposed to be professional endurance athletes?
- Response: Yes, the athletes belong to multidisciplinar Club called “Asociación Espeleológica Velezana”.
Comment 4.- 154: Are you meaning the root mean square of successive differences between R-R intervals?
- Response: Yes, It provides the root mean sum of the successive differences between R – R intervals (rMSSD) data using photoplethysmography.
Comment 5.- 172: Authors wrote the word outside twice
- Response: Thank you for this observation. “outside” has been removed.
Comment 6.- 179: Authors must present some studies where a period of 60-s for HRV measurements has been demonstrated to give reliable information rather than measuring for 5 minutes.
- Response: We appreciate the suggestions of the reviewer. The reference of validation of the use of HRV4training (60 seconds) for measuring HRV has been added in line 179. As well, below you can find some recent articles that have been used this measurement.
Flatt, A. A., Esco, M. R., y Nakamura, F, Y. (2017). Individual heart rate variability responses to preseason training in high level female soccer players. Journal of Strength and Conditioning Research 31(2), 531-538. Doi: 10.1519/JSC.0000000000001482
Javaloyes A.; Sarabia J. M.; Lamberts, R. P.; Plews, D.; Moya-Ramon, M. Training Prescription Guided by Heart Rate Variability Vs. Block Periodization in Well-Trained Cyclists. J Strength and Conditioning Research 2020 34(6), 1511–1518. https://doi.org/10.1519/JSC.0000000000003337
Plews D.J.; Scott B.; Altini M.; Wood M.; Kilding A.E.; Laursen P.B. Comparison of Heart-Rate-Variability Recording With Smartphone Photoplethysmography, Polar H7 Chest Strap, and Electrocardiography. Int J Sports Physiol Perform 2017, 12(10), 1324–8. https://doi.org/10.1123/ijspp.2016-0668
Comment 7.- 189: do authors mean maximal aerobic speed rather than maximal speed?
- Response: The “maximal speed” outcome is referred to the maximal speed obtained in the treadmill test in m/s. It is not related to the maximal aerobic speed. It has been specified in the text.
Round 2
Reviewer 2 Report
I agree with the authors that they have prepared their work in accordance with the guidelines presented in the above mentioned works. However, in my opinion, publishing a non-validated research protocol without verifying its effectiveness, i.e. presenting research results, is pointless. According to the authors, the objectives of the proposed protocol are: i) to determine changes in the performance of high-level athletes after following an HRV-guided or a traditional training period, and ii) to determine differences in the athletes' performance after following both training protocols. Therefore, the objectives were not achieved because no changes in the performance of the subjects were identified. The publication of such an unverified protocol may result in subsequent authors (trainers) applying it and consequently not achieving the training objectives. Researchers and readers rather expect a protocol with the results of the research. Based on the results of the research they will know whether the proposed protocol is worth applying in practice. Once again, they suggest that researchers add the data with research results and publish the paper as an original paper.
Author Response
We understand the considerations provided by the reviewer. As the reviewer points out, the objectives are not achieved. However, this is not the purpose of a trial protocol. Following the SPIRIT statement, the trial protocols are made to provide guidance to individuals conducting the study, to serve as the basis for trial registration, and to facilitate study appraisal by participants and external reviewers, including research ethics committees/institutional review boards, funders, regulators, journal editors, and systematic reviewers. A well structured protocol, addressing important study elements, will help to enhance trial implementation in order to reduce trial amendments and improving their quality. In this regard, we have followed the SPIRIT statement, developed using the systematic, transparent methodology and broad consultation with 115 experts representing diverse stakeholders involved in the design, funding, conduct, review, and publication of trial protocols. It adheres to the ethical principles mandated by the 2008 Declaration of Helsinki, and encompasses the protocol items recommended by the International Conference on Harmonisation Good Clinical Practice E6 guidance. It also supports trial registration requirements from the World Health Organization and the International Committee of Medical Journal Editors. On the other hand, the objectives designed in this protocol are prepared to be answered after the protocol implementation, independently of the results obtained in the randomized controlled trial ( i) to determine changes in the performance of high-level athletes after following an HRV-guided or a traditional training period, and ii) to determine differences in the athletes' performance after following both training protocols).
Reviewer 3 Report
The methods are very well described. Anyway, I don't think that a protocol is suitable to be published in this journal.
Following the given protocol, there will be interesting findings and results but it can not be accepted in its current form.
Author Response
We don’t understand the reviewer comments because the reviewer accepts that the introduction provides sufficient background and includes all relevant references, the research design is appropriate, and the methods are very well described. These aspects are the basis of a trial protocol. Following the SPIRIT statement, the trial protocols are made to provide guidance to individuals conducting the study, to serve as the basis for trial registration, and to facilitate study appraisal by participants and external reviewers, including research ethics committees/institutional review boards, funders, regulators, journal editors, and systematic reviewers. A well structured protocol, addressing important study elements, will help to enhance trial implementation in order to reduce trial amendments and improving their quality. In this regard, we have followed the SPIRIT statement, developed using systematic, transparent methodology and broad consultation with 115 experts representing diverse stakeholders involved in the design, funding, conduct, review, and publication of trial protocols. It adheres to the ethical principles mandated by the 2008 Declaration of Helsinki, and encompasses the protocol items recommended by the International Conference on Harmonisation Good Clinical Practice E6 guidance. It also supports trial registration requirements from the World Health Organization and the International Committee of Medical Journal Editors.
In this way, we don’t understand why the reviewer considers that the protocol is not suitable to be published in this journal. Specially when there are other similar trial protocol published in it. Please, see: https://www.mdpi.com/1660-4601/17/8/2919/htm, https://www.mdpi.com/1660-4601/17/8/2831, https://www.mdpi.com/1660-4601/17/4/1167/htm. As the reviewer can see, we have also added the Discussion section in order to adapt the structure to the journal guidelines.